# Statistical Analysis of Nutrient Loads from the Mississippi-Atchafalaya River Basin (MARB) to the Gulf of Mexico

**Phyllis Okwan** [1,*]**, Yi Zhen** [2]**, Huan Feng** [3] **, Shinjae Yoo** [4] **, Murty S. Kambhampati** [2]**, Abreione Walker** [2]**, Shayne Boykin** [2]**, Joe Omojola** [2] **and Noel Blackburn** [5]

[1] Department of Mathematics and Physics, Southern University and A&M College, Baton Rouge, LA 70813, USA

[2] Department of Natural Sciences, Southern University at New Orleans, New Orleans, LA 70126, USA; YZhen@suno.edu (Y.Z.); MKambham@suno.edu (M.S.K.); AWal1210@my.suno.edu (A.W.); sboy0917@my.suno.edu (S.B.); JOmojola@suno.edu (J.O.)

[3] Department of Earth and Environmental Studies, Montclair State University, Montclair, NJ 07043, USA; fengh@montclair.edu

[4] Computational Science Initiative, Brookhaven National Laboratory, Upton, NY 11973, USA; sjyoo@bnl.gov

[5] Office of Educational Programs, Brookhaven National Laboratory, Upton, NY 11973, USA; blackburn@bnl.gov

[*] Correspondence: phyllis_okwan@subr.edu

**Abstract:** This study investigated the annual and seasonal variations in nutrient loads ($NO_2^- + NO_3^-$ and orthophosphate) delivered to the Gulf of Mexico from the Mississippi-Atchafalaya River Basin (MARB) and examined the water quality variations. The results indicate that (1) annually, the mean $NO_2^- + NO_3^-$ and orthophosphate loads showed a steady increase during 1996–1999, a persistent level during 2000–2007, and a moderate increase during 2008–2016; (2) seasonally, $NO_2^- + NO_3^-$ and orthophosphate in MARB in spring and summer were higher than those in autumn and winter. Analysis of variance (ANOVA) identified highly significant differences among seasonal loads; and (3) the median value of $NO_2^- + NO_3^-$ in normal weather conditions were higher than that during and right after the hurricanes, while the median value of orthophosphate loads in normal weather conditions was higher than that during the hurricanes, but higher than that right after hurricanes. The two-sample t-test indicates a significant difference ($p < 0.046$) in orthophosphate loads before and after Hurricane Katrina. Moreover, it is found that there is a significant ($p < 0.01$) increase in nutrient loads during normal weather conditions. The results indicate that hurricane seasons can significantly influence the nutrient loads from the MARB to the Gulf of Mexico.

**Keywords:** Gulf of Mexico; nutrient loads; seasonal variations; water quality

## 1. Introduction

The "hypoxic zone" refers to a specific water area where the dissolved oxygen concentration is less than 2 mg/L, and to hypoxia associated with excess nutrient input, such as nitrogen and phosphorus, which can yield serious environmental, economic, and social impact. A variety of sources, which are related to economic activities and population growth, can contribute to the elevated nutrient concentrations in rivers (e.g., agricultural activities, industrial point-discharge, urban runoff, natural deposition, and domestic waste, such as household cleaning products; [1–6]. Water quality from the Mississippi-Atchafalaya river basin to the Northern Gulf of Mexico has been improved since the enactment of the 1972 Clean Water Act (CWA) and other regulations by the Environmental Protection

Agency (EPA) [1]. According to a previous study [1], agriculture has been considered as one of the pollution sources of phosphates and nitrates. In urban areas, domestic fertilizer and detergent use in the residential areas along the rivers or in the watersheds can also discharge nitrate, nitrite, ammonium, and phosphate directly into rivers or through municipal sewage run-offs, as a result of urbanization and population increase [7]. High nutrient concentrations in water have caused major problems to the quality of rivers and coastal waters [1,8–10]. Phosphorus plays a major role in productivity of hypoxia in the Gulf of Mexico [11,12]. It is also noted that nitrate is one of the top groundwater contaminants in rural areas [13–15].

The development of a hypoxic zone in the Gulf of Mexico has been associated with several factors, of which excess nutrient is agreed as one of the major factors [16–22]. In response to the severe situation of the hypoxic zone in the Gulf of Mexico, the Hypoxia Task Force (HTF) was established in 1997 and its goal is to reduce excess nitrogen and phosphorus input in the Mississippi-Atchafalaya River Basin (MARB). The action plan to Congress was delivered by HTF in 2001 entitled Action Plan for Reducing and Controlling Hypoxia in the Northern Gulf of Mexico (https://www.epa.gov/ms-htf/hypoxia-task-force-2001-action-plan). The action plan provided a national strategy to reduce the frequency, duration, size, and degree of the oxygen depletion of the hypoxic zone in the northern Gulf of Mexico. In order to pinpoint the location and sources origination, a Spatially Referenced Regression on Watershed (SPARROW) was constructed for the MARB in 1992. The SPARROW model of the MARB showed that the states of Illinois, Iowa, Indiana, Missouri, Arkansas, Kentucky, Tennessee, Ohio, and Mississippi make up one-third of the 32-state MARB area and contribute more than 75% of nitrogen and phosphorus to the Gulf. Agricultural non-point sources contributed more than 70% of the nitrogen and phosphorus delivered to the Gulf, versus only about 9% to 12% from urban sources. Wild animal manure on pasture and rangelands contributed nearly as much phosphorus as cultivated crops, 37% versus 43%, indicating the wastes of unconfined animals is a much larger source of phosphorus in the MARB than previously [11,12,23]. Atmospheric condition is also important, accounting for 16% of nitrogen. Corn and soybean cultivation was the largest contributor of nitrogen to the Gulf—66% of nitrogen originated from cultivated crops, mostly corn and soybean crops, with animal grazing and manure contributing only about 5% [11,23–26]. The new estimations from SPARROW for the MARB indicate that the Corn Belt (centered over Iowa and Indiana) yielded the highest nitrogen [27,28]. Locations producing high phosphorus yields were scattered throughout the MARB. Agricultural activities (fertilizer, manure, and fixation) are the dominant sources of nitrogen and phosphorus. Area ranking for nitrogen has negligible changes, while that for phosphorus exhibit a striking variation. Dominant sources of phosphorus were identified as crop and animal agriculture [29,30] and major wastewater treatment plants (WWTP) (https://www.epa.gov/npdes/municipal-wastewater). Therefore, it is of importance to identify and quantify the effects of these contaminants from rivers. In order to quantify the relationship between excess nutrients and development of hypoxic zone in the northern Gulf of Mexico, this study focuses on the nutrient loads from the Mississippi-Atchafalaya River Basin and aims to reveal the patterns of annual variations and monthly variations of nutrient loads, as well as variations under extreme weather conditions. The purpose of this study is to estimate the nutrient loads using available data and to investigate the impact of hurricanes on nutrient loads.

## 2. Materials and Methods

### 2.1. Study Area

The Mississippi River Basin is the largest river basin in North America and the third largest river basin in the world [20]. The Mississippi-Atchafalaya River Basin (MARB) encompasses both the Mississippi and Atchafalaya River Basins. The Mississippi River covers approximately 41% of the conterminous US and has the seventh largest water discharge and suspended loads of world waters [31,32]. According to the United States' Environmental Protection Agency (EPA), the Mississippi River is approximately 3782 km in length, receiving water flow from 32 states, and meets up with

its distributary, the Atchafalaya River. The Mississippi River either borders or passes through ten states starting from Minnesota through Wisconsin, Iowa, Illinois, Missouri, Kentucky, Tennessee, Arkansas, Mississippi, and Louisiana into the Gulf of Mexico. The Atchafalaya River is located within the Mississippi River delta plain in south Louisiana. The Atchafalaya River is 137 miles long (220 km), and a distributary of the Mississippi River and the Red River in south central Louisiana [33,34]. The Mississippi-Atchafalaya River Basin (MARB) is shown in Figure 1. The Mississippi River empties into the Gulf of Mexico and the Atchafalaya River is a distributary of the Mississippi River and Red River in south central Louisiana in the United States. The Atchafalaya River meets up with Mississippi River before reaching the Gulf of Mexico.

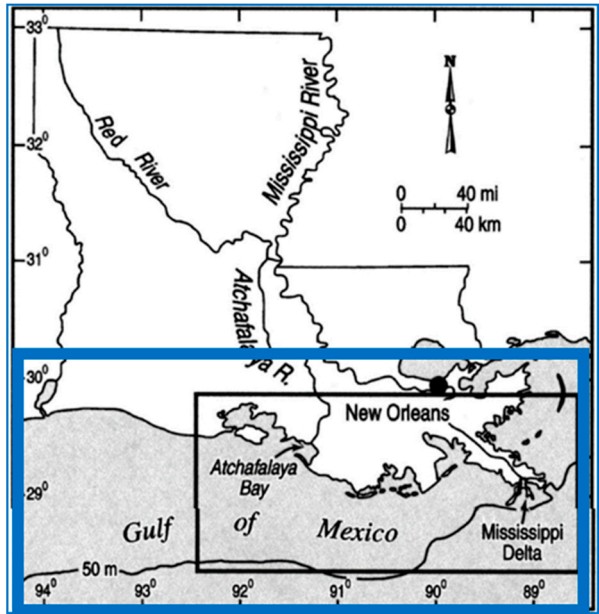

**Figure 1.** Map showing the Mississippi-Atchafalaya River Basin (MARB) from a Google map with line color modification.

*2.2. Materials*

The data used in this study, including the annual total loads of inorganic nitrogen and orthophosphate, were collected from the United States Geological Survey (USGS) archive data (https://toxics.usgs.gov/hypoxia/mississippi/flux_ests/delivery/index.html). The data were analyzed for annual and seasonal variation patterns of $NO_2^- + NO_3^-$ and orthophosphate loads from the Mississippi-Atchafalaya River Basin that were delivered to the Gulf of Mexico during 1996–2016. Nutrient load data used for analysis were the sample data of total inorganic nitrogen ($NO_2^- + NO_3^-$) and total orthophosphate. The load estimator (LOADEST)data adequately reflect the effect of nutrient loads on water quality from its source to the gulf mouth, particularly with respect to the influence of agricultural activities on the Mississippi-Atchafalaya River Basin on hypoxia in the northern Gulf of Mexico. The composite data were used for autoregressive integrated moving average (ARIMA) analysis.

*2.3. Data Processing and Statistical Analysis*

A long-term time series analysis was performed on the USGS nutrient load dataset, which were estimated by the adjusted maximum likelihood estimation (AMLE) method using the LOADEST program and by the composite method. The annual and monthly loads were used to reveal annual and seasonal characteristics of the loads, respectively. Specifically, annual loads of inorganic nitrogen ($NO_2^- + NO_3^-$) and orthophosphate from 1996 to 2016 were used to study the nutrient variations. Monthly net loads of inorganic nitrogen ($NO_2^- + NO_3^-$) and orthophosphate during 1996 to 2016 were used to examine the nutrient seasonal variations. Statistical analysis was carried out to examine the rise,

peak, and impact of storm flow conditions and seasonal conditions on nutrients loads. The seasons were defined as winter (December–February), spring (March–May), summer (June–August), and autumn (September–November) [35,36]. Two-sample t-tests and analysis of variance (ANOVA) were applied to identify the significant variations in the mean nutrient loads among different seasons and weather conditions. A forecasting $NO_2^- + NO_3^-$ and orthophosphate model of net nutrient loads from MARB to the Gulf of Mexico was established by the autoregressive integrated moving average (ARIMA) method. The mean monthly nutrient loads in the MARB delivered to the Gulf of Mexico from October 1995 to September 2016 were used for constructing the predictive autoregressive integrated moving average (ARIMA) model. The order of autoregressive terms (p) were estimated by a plot of the partial autocorrelation function (PACF), and the order of the moving-average terms (q) were estimated by the plot of the autocorrelation function (ACF). The ARIMA models are all possible combinations of p and q. The final ARIMA model was selected based on the Akaike Information Criterion (AIC). The total number of observations were 252, of which 180 observations from October 1995 to September 2010 were used as the training data while the remaining 72 observations from October 2010 to September 2016 were used as testing data. In this study, weather conditions and annual data were divided into three categories representing "no hurricane", "hurricane", and "after hurricane". The category of no hurricane means normal conditions under which calculations exclude loads in a hurricane year. In addition, Hurricane Katrina in 2005 was treated as cutting point to separate the hurricane weather conditions from the normal weather condition in order to evaluate the effect of hurricane condition on variations in nutrient loads.

## 3. Results and Discussion

### 3.1. Annual Variation

The composite method load estimate was reported when there were 10 or more nutrient concentration measurements for that year. The nutrient loads were calculated using a 5-year moving calibration method to investigate the annual and seasonal variations in the nutrients loads to the Gulf of Mexico delivered from the Mississippi-Atchafalaya River Basin. Method comparison between the LOADEST and composite methods for the nutrient loads was made and the results are shown in Figure 2. The two-sample t-test showed that there is no statistically significant difference ($p > 0.05$) between the two methods for the nutrient loads. The annual nutrient loads estimated by the LOADEST method show that the nutrient loads from 1996 to 2016 were 892,286 t/yr with a range of 671,000 t/yr for $NO_2^- + NO_3^-$, and 44,324 t/yr with a range of 39,700 t/yr for orthophosphate, respectively. In the meantime, the annual nutrient loads given by the composite method for the same time period were 898,143 t/yr with a range of 755,000 t/yr for $NO_2^- + NO_3^-$, and 44,319 t/yr with a range of 41,100 t/yr for orthophosphate, respectively. It is found that, for the LOADEST method, the $NO_2^- + NO_3^-$ loads increase from 827,000 t/yr in 1996 to 1,110,000 t/yr in 1999, and then declined to 539,000 t/yr and remained constant at around 800,000 t/yr during 2001 and 2007, then increased to 1,210,000 t/yr and persisted at around 1,000,000 t/yr during 2008 and 2011, and then decreased to 571,000 t/yr in 2012 and increased again to 1,100,000 t/yr in 2016. The annual predicted nutrient loads indicate that the peak $NO_2^- + NO_3^-$ load appeared in 2008, showing an increasing trend in the last two decades. The mean $NO_2^- + NO_3^-$ loads have been increasing since 1996 but the rate has declined in recent years. $NO_2^- + NO_3^-$ loads exhibit slightly increasing trends over the last two decades. The mean orthophosphate increased from 42,700 t/yr in 1996 to 63,200 t/yr in 2008. The peak orthophosphate loads appeared in 2008, showing an increasing trend over the last two decades. The increasing rate weakened by 39% in recent years. The nutrient loads before and after hurricane Katrina in 2005 are shown in Figure 3.

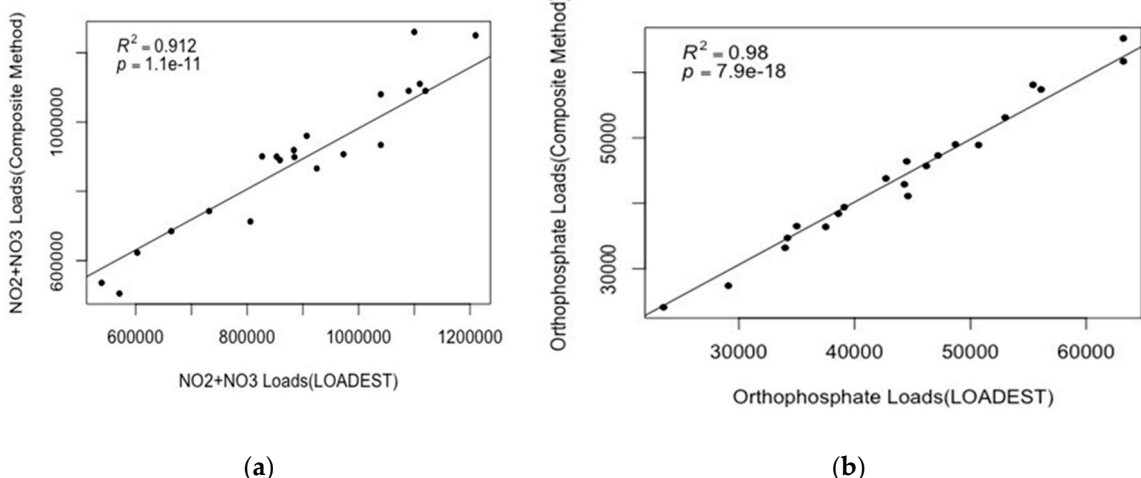

**Figure 2.** Comparison of the annual nutrient loads estimated by the LOADEST and composite methods for (**a**) $NO_2^- + NO_3^-$ and (**b**) orthophosphate during the period of 1996–2016.

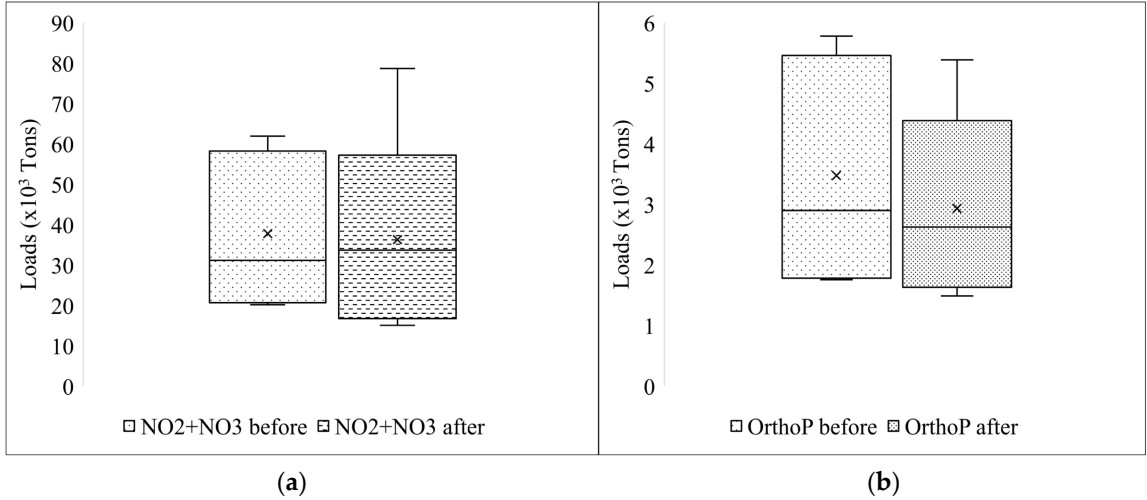

**Figure 3.** Comparison of nutrient loads before and after Hurricane Katrina in 2005 for (**a**) $NO_2^- + NO_3^-$ and (**b**) orthophosphate. There is no statistically significant difference ($p > 0.05$) between before and after Hurricane Katrina in 2005 for $NO_2^- + NO_3^-$. However, there is a statistically significant difference ($p < 0.05$) between before and after Hurricane Katrina in 2005 for orthophosphate.

A two-sample t-test was performed to compare the mean $NO_2^- + NO_3^-$ and orthophosphate loads before and after Hurricane Katrina in 2005. The t-test results indicate a statistically significant increase in the mean orthophosphate load after Hurricane Katrina ($p < 0.05$). The mean load increased from 39,888.89 t/yr to 48,800.00 t/yr. A Wilcoxon rank-sum test was performed to detect significant difference in monthly $NO_2^- + NO_3^-$ and orthophosphate loads before and after a hurricane and the results indicated that there was no statistically significant difference ($p > 0.05$) between a month before and after the hurricane for both $NO_2^- + NO_3^-$ and orthophosphate loads, respectively.

### 3.2. Seasonal Variation

Variations in nutrient loads can be significantly affected by meteorological conditions, biomass productivity, and land-use activities, of which the severe weather involving strong winds and waves also yield a great impact [37,38]. The seasonal variations in the mean loads are summarized in Figure 4, which show that the mean loads of spring and summer are higher than those in autumn and winter.

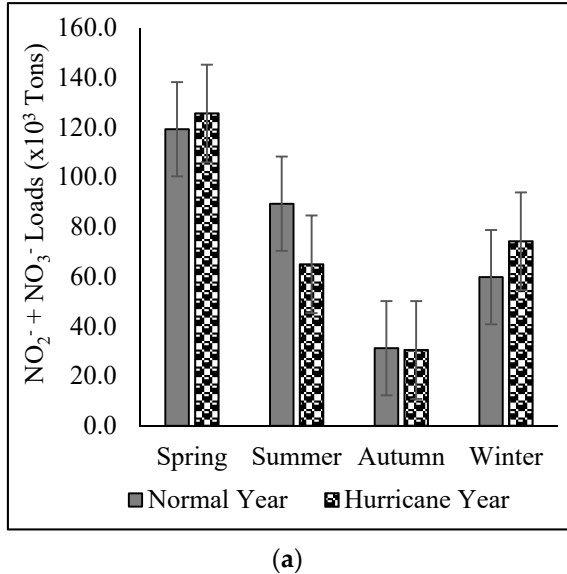 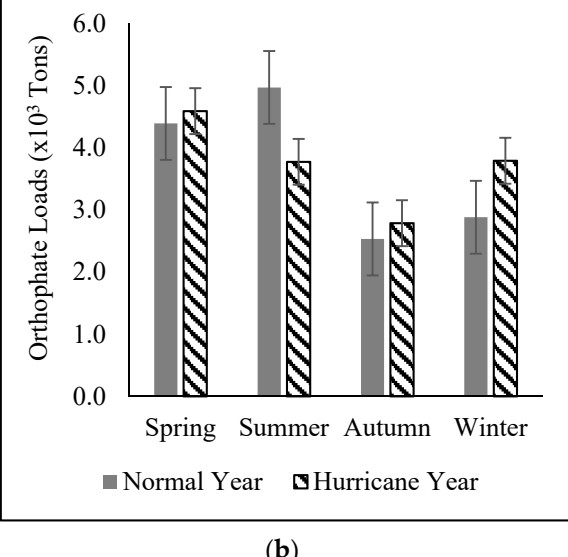

(a)　　　　　　　　　　　　　　　　　(b)

**Figure 4.** Comparison of seasonal variations of (**a**) $NO_2^- + NO_3^-$ and (**b**) orthophosphate under normal weather and hurricane conditions. Note: Normal weather are years without any storm, and hurricane conditions are hurricane occurrence years.

One-way ANOVA was conducted to identify the significant difference in seasonal $NO_2^- + NO_3^-$ and orthophosphate loads. The ANOVA results indicate that there is significant difference existing in seasonal $NO_2^- + NO_3^-$ load ($p < 0.01$). A multiple comparison Tukey's test indicates that winter load is significantly different from spring and autumn loads ($p < 0.05$); spring load is significantly different from summer load and autumn load ($p < 0.05$); and summer load is significantly different from autumn load ($p < 0.05$). Among the four seasonal periods, the mean $NO_2^- + NO_3^-$ load is in the order of Spring > Summer > Winter > Autumn. For orthophosphate load, statistical results indicate a significant difference ($p < 0.05$) in nutrient load among four seasons. In contrast, the mean orthophosphate load is in the order of Summer > Spring > Winter > Autumn. The load in the summer season made up the largest proportion (33.6%), followed by spring (29.7%), winter (19.5%), and autumn (17.1%) under normal weather conditions. The seasonal cycle that mostly reflects freshwater discharge of $NO_2^- + NO_3^-$ loads in Mississippi river is summer. Highest loads of nitrogen usually occur in May/June, while loads are lowest in September/October [39].

Under hurricane weather conditions, however, spring season made up the largest proportion (30.7%), followed by winter (25.4%), summer (25.3%), and autumn (18.7%).

During 1996–2016, hurricane years were 2002 (Hurricane Lili), 2005 (Hurricane Cindy, Hurricane Dennis, Hurricane Katrina, Hurricane Rita), 2007 (Hurricane Humberto), 2008 (Hurricane Gustav, Hurricane Ike), 2009 (Hurricane Ida), and 2012 (Hurricane Isaac). The category of "no hurricane" includes years 1996, 1997, 1998, 1999, 2000, and 2001; hurricane category includes years 2002, 2005, 2007, 2008, 2009, and 2012; after hurricane years include 2003, 2004, 2006, 2010, 2011, 2013, 2014, 2015, and 2016. The nutrient loads under different weather conditions are shown in Figure 5.

ANOVA was performed on these three categories of $NO_2^- + NO_3^-$ and orthophosphate loads, respectively, to detect if there is statistically significant difference among the mean loads of the three groups in order to assess the impact of extreme weather conditions on variations of nutrient loads. The ANOVA results indicated there is no statistically significant difference ($p > 0.05$) among the three categories of weather condition for both $NO_2^- + NO_3^-$ and orthophosphate loads. The mean loads under different weather conditions indicate that $NO_2^- + NO_3^-$ load under hurricane conditions was about 1.24 times greater than that under normal weather conditions. The relative contributions of seasonal loads to the total loads of $NO_2^- + NO_3^-$ under normal weather and hurricane conditions

varied. It is interesting to find that hurricane flow seems to be associated with an increase in nutrient loads during winter.

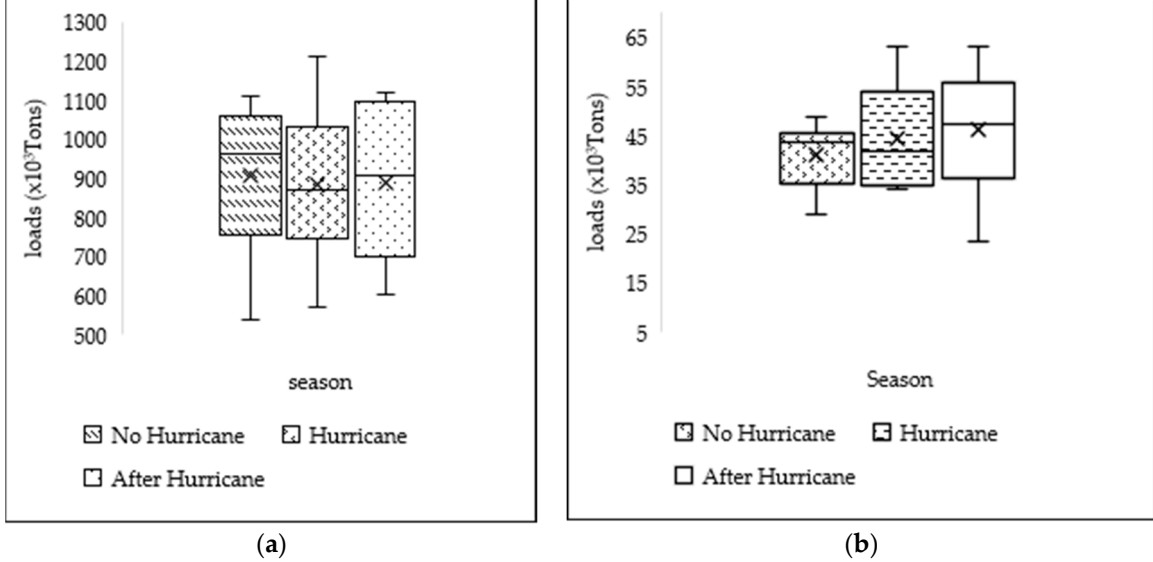

**Figure 5.** Comparison of variations in (**a**) $NO_2^- + NO_3^-$ load and (**b**) orthophosphate load under normal weather, during hurricane, and after hurricane conditions.

### 3.3. Nutrient Load Prediction

An autoregressive integrated moving average (ARIMA) model was constructed for the nutrient loads forecast. The Dickey–Fuller test results showed that both $NO_2^- + NO_3^-$ and orthophosphate annual loads from 1996 to 2016 were not stationary data series with $p > 0.05$ in both cases. Therefore, the ARIMA method is not suitable for the annual dataset. Nevertheless, an augmented Dickey–Fuller Test on the monthly $NO_2^- + NO_3^-$ loads did not detect any non-stationarity data in the data set ($p < 0.05$). The decomposition analysis suggests a seasonal component for the model. The monthly observation of $NO_2^- + NO_3^-$ load is shown in Figure 6.

The autocorrelation function (ACF) and the partial autocorrelation function (PACF) for the monthly data set were computed. As shown in Figure 6, the ACF plot suggests a possibility of MA (1) and MA (2) terms and the PACF plot indicates AR (1), AR (2), AR (3), and AR (4) terms for the model, respectively. Based on the AIC value of the models, ARIMA $(1,0,2) \times (0,1,1)_{12}$ is selected as the final model which is expressed as follows:

$$(1 - \phi_1 B)(1 - B^S)(X_t - \mu) = (1 + \theta_1 B + \theta_2 B^2)(1 + \theta_1^s B^{1,S})w_t,$$

where $B$ is the backshift operator, $B^S$ is the seasonal backshift operator, $\phi_1 = 0.8013$ is the coefficient of AR (1) term, $\theta_1^S = -0.960$ is the coefficient of the seasonal MA (1) term, and $\theta_1 = -0.1799$ and $\theta_2 = -0.1790$ are coefficients of the MA (1) and MA (2) terms (Table 1).

The residuals of ARIMA $(1,0,2) \times (0,1,1)_{12}$ for the $NO_2^- + NO_3^-$ loads are shown in Figure 6. The residuals plot indicates that the model captures the time series attributes of the observations. The monthly $NO_2^- + NO_3^-$ forecasting load is shown in Figure 6 along with observations in the testing dataset.

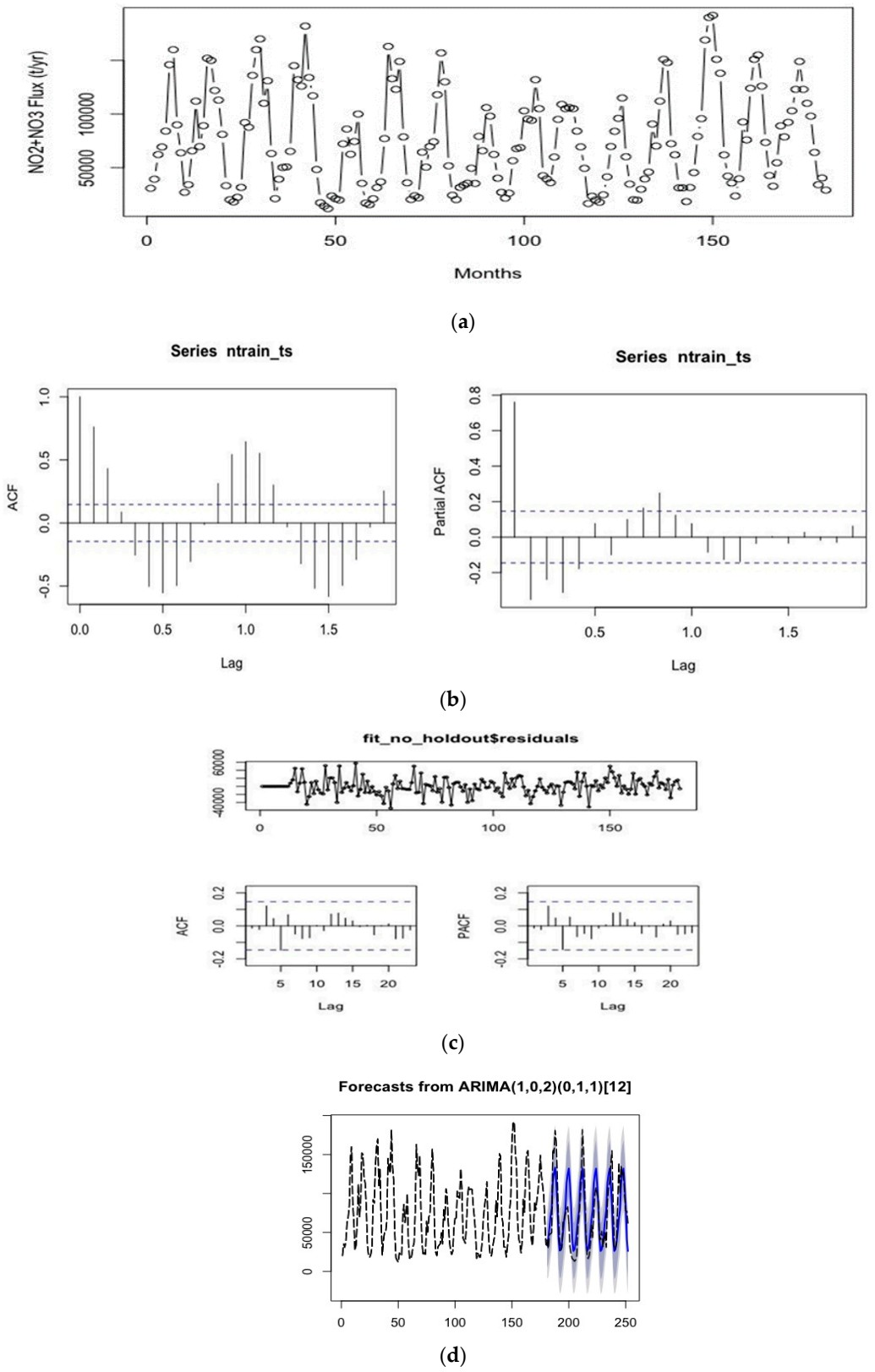

**Figure 6.** Results of the autoregression integrated moving average (ARIMA) analysis of $NO_2^- + NO_3^-$ data. (**a**) Monthly observations of the $NO_2^- + NO_3^-$ load in the Mississippi River; (**b**) ACF and PACF plots of the training data; (**c**) residual plot of ARIMA $(1,0,2) \times (0,1,1)_{12}$; and (**d**) model forecast for mean monthly nutrient $NO_2^- + NO_3^-$ loads from October 2010 through September 2016.

**Table 1.** Coefficient estimations of ARIMA (1,0,2) (0,1,1)$_{12}$.

| Coefficients | ar1 | ma1 | ma2 | sma1 |
|---|---|---|---|---|
| ARIMA (1,0,2)(0,1,1)$_{12}$ | 0.8013 | −0.1799 | −0.179 | −0.96 |
| s.e | 0.0913 | 0.1191 | 0.0981 | 0.226 |

Sigma$^2$ estimated as 452750783, log likelihood = −1925.97, AIC = 3861.93.

For the monthly data of the orthophosphate loads, the ARIMA model is established on the logarithm-transformed data set. The augmented Dickey-Fuller test confirms the stationarity of the data set ($p < 0.05$). The decomposition analysis suggests a seasonal component for the model. The monthly observation of orthophosphate load is shown in Figure 7.

The ACF and PACF of logarithm-transformed data were computed and shown in Figure 7. In Figure 7, it can be seen that the ACF and PACF suggest the possibility of MA (1), MA (2), and MA (3) terms, as well as an AR (1) term for the model, respectively. Based on the AIC value of the models, the final model is defined as ARIMA (1,0,0) $\times (0,1,1)_{12}$:

$$(1 - \phi_1 B)\left(1 - B^S\right)(ln(X_t) - \mu) = \left(1 + \theta_1^S B^{1,S}\right)w_t,$$

where $B$ is the backshift operator, $B^S$ is the seasonal backshift operator, $\phi_1 = 0.7213$ is the coefficient of AR (1), and $\theta_1^S = -0.8996$ is the coefficient of the seasonal MA (1) term (Table 2).

**Table 2.** Coefficient estimations of ARIMA (1,0,0) (0,1,1)$_{12}$.

| Coefficients | ar1 | sma1 |
|---|---|---|
| ARIMA(1,0,0)(0,1,1)$_{12}$ | 0.7213 | −0.8996 |
| s.e | 0.0535 | 0.1021 |

Sigma2 estimated as 0.08145, log likelihood = 37.76, AIC = 81.52.

The residuals of ARIMA (1,0,0) $\times (0,1,1)_{12}$ are shown in Figure 7. From Figure 7, it can be seen that the model captured the characteristics of the time series observations of the orthophosphate loads. The predicted values are logarithm transformed. The forecasting values of logarithm-transformed loads are presented along with the logarithm of the original observations in the testing dataset in Figure 7.

The mean annual nutrient loads ($10^3$ t/yr) observed during 1996–2016 and predicted for 2017–2022 are summarized in Table 3.

Statistically, there is no significant difference ($p > 0.05$) between the observed and the predicted dataset.

**Table 3.** Comparison of observed (1996–2016) and predicted (2017–2022) annual mean ($NO_2^-$ + $NO_3^-$) and orthophosphate loads ($10^3$ t/yr).

| Nutrient Loads | Observation | Prediction |
|---|---|---|
| | 1996–2016 | 2017–2022 |
| $NO_2^-$ + $NO_3^-$ | 892 ± 192 | 1100 ± 22 |
| Orthophosphate | 44.3 ± 10.5 | 41.7 ± 5.25 |

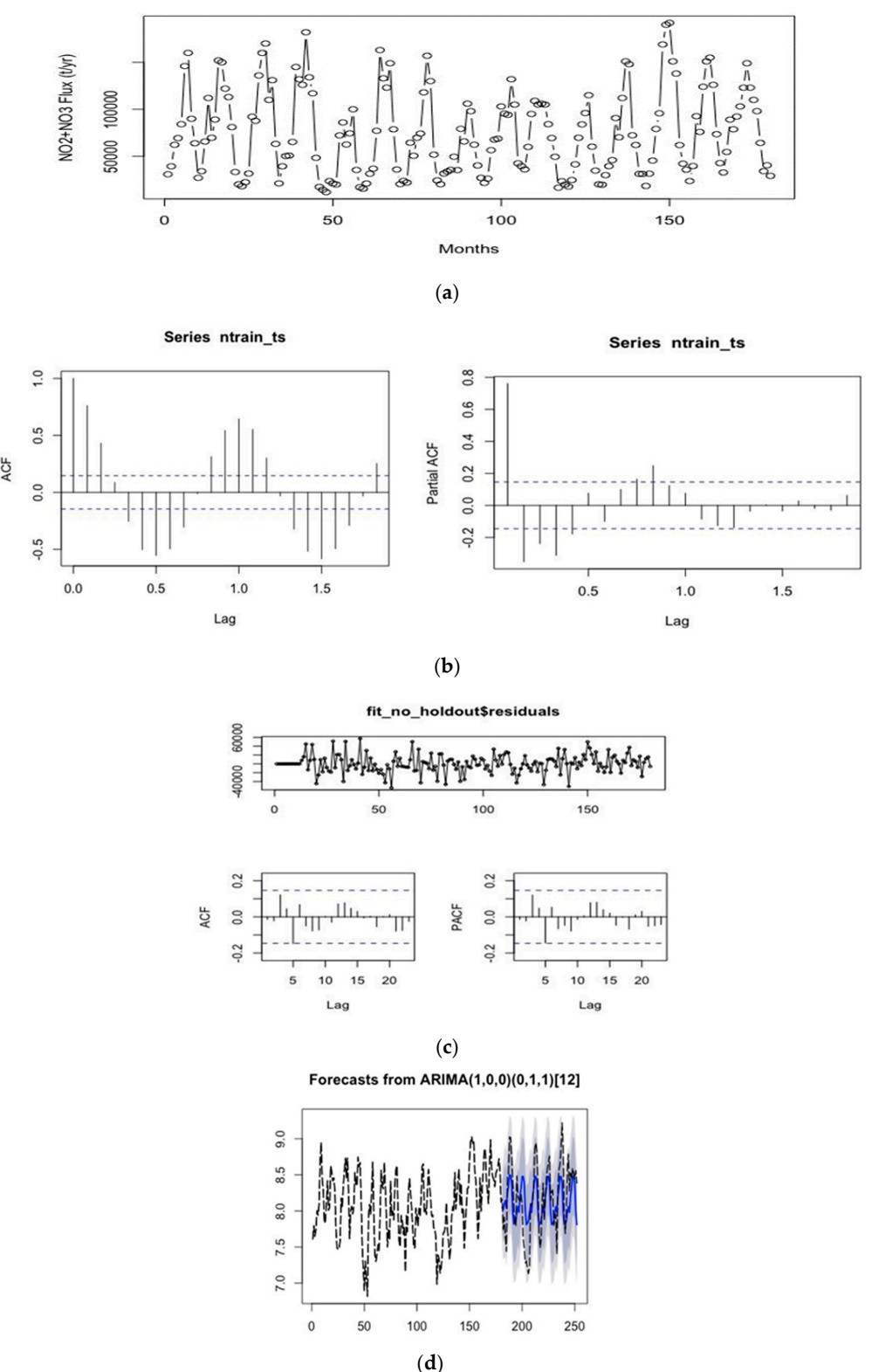

**Figure 7.** Results of the autoregression integrated moving average (ARIMA) analysis of orthophosphate data. (**a**) Monthly observations of orthophosphate concentration expressed on a linear scale and logarithmic transformation; (**b**) ACF and PACF plots of the training data; (**c**) the residuals plot of ARIMA $(1,0,0) \times (0,1,1)_{12}$; and (**d**) mean monthly forecasts of orthophosphate load from October 2010 through September 2016 on a logarithmic scale (the blue line is a predictive line).

The forecasts of monthly mean loads of $NO_2^- + NO_3^-$ from 2017 to 2022 are obtained from ARIMA $(1,0,2) \times (0,1,1)_{12}$. Then, the annual predicted mean loads of $NO_2^- + NO_3^-$ are obtained from the summation of the predicted values of twelve months. The prediction for annual mean $NO_2^- + NO_3^-$ loads is 1100.3 ± 22.0 t/yr, which is 23% higher than the actual observations of 892.3 ± 192.1 t/yr. The standard deviation of the predicted loads is 89% lower than that of the observed loads. The monthly mean loads of orthophosphate loads from 2017 to 2022 are predicted from ARIMA $(1,0,0) \times (0,1,1)_{12}$. The annual predicted mean loads of the orthophosphate loads are obtained by addition of the predicted loads of twelve months. The prediction for annual mean orthophosphate loads is 41.7 ± 5.25 t/yr, which is 5.8% lower than the actual observations of 44.3 ± 10.5 t/yr. The two-sample t-test for observed and predicted annual mean nutrient loads indicate that there is no statistically significant difference ($p > 0.05$). Therefore, this model has significance to apply in other regions with similar settings.

## 4. Conclusions

The results from this study suggest that the mean orthophosphate load is more sensitive to changes in weather conditions than mean $NO_2^- + NO_3^-$ load in the Mississippi-Atchafalaya River Basin. The monthly observations of nutrient loads exhibit obvious seasonal variation. The loads showed a trend of being maximum in spring and minimum in autumn during the hurricane seasons. Time series analysis indicate that the $NO_2^- + NO_3^-$ load in the Mississippi-Atchafalaya River Basin has a significant seasonal attribute, strong correlations among annual and monthly loads, as well as considerable sensitivity to weather conditions, which is consistent [40]. Overall, the results suggest that hurricane seasons significantly influence the nutrient loads from the Mississippi-Atchafalaya River Basin (MARB) to the Gulf of Mexico.

**Author Contributions:** Conceptualization, M.S.K., J.O., P.O., Y.Z., H.F., S.Y., and N.B.; Methodology, S.Y., H.F., P.O., Y.Z., A.W., and S.B.; Software, P.O., Y.Z., and H.F; Validation, H.F., M.S.K., Y.Z., P.O., and J.O.; Formal analysis, P.O., Y.Z., A.W., and S.B.; Investigation resources, M.S.K., J.O., and P.O.; Data curation, H.F., S.Y., P.O., and Y.Z.; Writing-original draft preparation, P.O., Y.Z., and H.F.; Writing—review and editing, H.F., P.O., M.S.K., and J.O., Visualization, Y.Z., P.O., S.B., A.W.; Supervision, H.F., S.Y., P.O., Y.Z., and N.B., Project administration, H.F. and S.Y.; Funding acquisition, M.S.K., J.O., and P.O. All authors have read and agreed to the published version of the manuscript.

**Funding:** The US Department of Education—Minority Science and Engineering Improvement Program—Capacity Competitiveness Enhancement Model (MSEIP-CCEM grant; P120A160047) at Southern University at New Orleans.

**Acknowledgments:** This study was supported by and the 2019 College Research Teams Program (CRTP) at Brookhaven National Laboratory. We also thank the staff members in the Office of Education Programs and the Computational Science Initiatives at Brookhaven National Laboratory for their support and assistance in this research.

**Conflicts of Interest:** The authors declare no conflict of interest.

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
