# Peer review of "Statistical Analysis of Nutrient Loads from the Mississippi-Atchafalaya River Basin (MARB) to the Gulf of Mexico"

_environments, doi:10.3390/environments7010008_

Round 1

Reviewer 1 Report

Investigation of trends in loads is very often investigation of trends in water flow (load=concentration*flow). There is sense to compare them in investigation.

Author Response

Investigation of trends in loads is very often investigation of trends in water flow (load=concentration*flow). There is sense to compare them in investigation.

Response: Thank you for your encouragement.

Reviewer 2 Report

The manuscript describes an interesting statistical analysis of long time series of nutrient loads from the Mississippi-Atchafalaya river basin (I suppose to the outlet). They calculated the nutrient loads of long time series of NO2+NO3 and PO4 in the period 1996-2016 using the LOADEST program, then they applied statistical analysis to compare the time series before and after the Hurricane Catrina, the seasonal variations using the ANOVA test and  they performed the ARIMA model for the prediction and analysis of nutrient loads.

Albeit the topic is interesting, introducing the analysis of the loads during an extreme meteorological data like Hurricane Catrina, the description of the methodology is very poor and not well organized.

In my opinion, the main issue concerns the fact that the description of input data is very poor as well as the statistical methods that are mainly reported in the results section. Figures have a course resolution and are often divided inside a template of table (???).  More information regarding the Hurricane Catrina should be reported (precipitation, increase of flow…etc.).  I have also some specific questions, than maybe can help to improve the manuscript:

-why have you analyzed the sum of NO2 and NO3 and not just the Nitrates (maybe is the main component of TN in the Mississipi River Basin… see Silver et al., 2010 DOI: 10.2134/jeq2010.0115 )?

-why you didn’t’ analyze the long times series of concentrations instead of the loads that are affected by the streamflow variability and uncertainty? (see Sprague et al., 2011, doi: 10.1021/es201221s)

-I would like to understand better the ARIMA stages i) model selection, ii)  parameter estimation and iii) diagnostic checking. These important stages are not well explained and should be reported in the methodology

-The prediction of the loads should be compared with the observation to understand the model performance

Other specific comments:

-L.33 2 space mg/L

-L.38 Add citation: Malago' et al. 2019 (Malagó, A.; Bouraoui, F.; Grizzetti, B.; De Roo, A. Modelling nutrient fluxes into the Mediterranean Sea.J. Hydrol. Reg. Stud. 2019, 22, 100592)

-L.64 Noncaptured? Uncaptured…but what do you mean? Wild?

-L.64-68 add references

-L.80 Add more details about the main goals of the work and main methods applied….

-L.119 explain why you used the sum of No2 and No3. Add data entries and period of each variable in section 2.2. Here you should describe the observed time series not in the results (Figure 6 a and 7 a)…

-Section 2.3. what do you mean with Net loads?  Rewrite this section adding

Method of calculation of loads Explain why you decide to calculate loads and not use directly concentrations Explain how have you calculated the trends as reported in line 170 I suggest to create a figure with the flow chart of work.

-L.151-153 is method

-L.223-231 is method

-L.241 Some explanations of that?

-L.243-249 all the tests have to be described in the method!!

-L.271 and L. 331 all the equations should be explained in the method

-the conclusions are very poor and not well supported by the results. I suggest to add a discussion section.

About Figures

-Fig.1 delete part a) and show the position of monitoring point.

-Fig.2 improve the figure and use consistent terminology OrthoP or PO4? Tons or t/yr…

Author Response

The manuscript describes an interesting statistical analysis of long time series of nutrient loads from the Mississippi-Atchafalaya river basin (I suppose to the outlet). They calculated the nutrient loads of long time series of NO2+NO3 and PO4 in the period 1996-2016 using the LOADEST program, then they applied statistical analysis to compare the time series before and after the Hurricane Catrina, the seasonal variations using the ANOVA test and  they performed the ARIMA model for the prediction and analysis of nutrient loads.

Response:  Thank you for the summary.

Albeit the topic is interesting, introducing the analysis of the loads during an extreme meteorological data like Hurricane Catrina, the description of the methodology is very poor and not well organized.

Response: Thank you for your comments.  However, could you give more detail, specific comments so that we can improve the description in a better way?

In my opinion, the main issue concerns the fact that the description of input data is very poor as well as the statistical methods that are mainly reported in the results section. Figures have a course resolution and are often divided inside a template of table (???). 

Response: We revised the description of the data input and gave more explanation of the statistical methods.  The quality of some figures has been improved.

More information regarding the Hurricane Catrina should be reported (precipitation, increase of flow…etc.).  I have also some specific questions, than maybe can help to improve the manuscript:

Response: We thought that the river flow before and after the Hurricane Katrina is reported in the original manuscript.  Could you be more specific?  Precipitation is related to the river flow. We did not intend to use the precipitation specifically in this study.

-why have you analyzed the sum of NO2 and NO3 and not just the Nitrates (maybe is the main component of TN in the Mississipi River Basin… see Silver et al., 2010 DOI: 10.2134/jeq2010.0115 )?

Response: In the original data file from U.S. Geological Survey report, the loads of NO2 and NO3 were added together as (NO2 + NO3) and then reported. Therefore, we do not have separate NO2 and NO3 data and have to use the combined data.  Thank you for recommending the publication by Tukamushaba David Silver et al. (2010).  This is a useful reference to our manuscript.

-why you didn’t’ analyze the long times series of concentrations instead of the loads that are affected by the streamflow variability and uncertainty? (see Sprague et al., 2011, doi: 10.1021/es201221s)

Response: We received only nutrient load information from the U.S. Geological Survey report.  Therefore, we do not have nutrient concentration data for the study area from the data report. As this study focuses on the nutrient loads, we feel that it should be fine to use the load data only.  Thank you for recommending the publication by Sprague et al. (2011).  We cited it in our study.

-I would like to understand better the ARIMA stages i) model selection, ii)  parameter estimation and iii) diagnostic checking. These important stages are not well explained and should be reported in the methodology

Response: Thank you for your comments and suggestions.  We have revised the section accordingly. The sentences, “The order of autoregressive terms (p) are estimated by plot of partial autocorrelation function (PACF) and the order of moving-average terms (q) are estimated by plot of autocorrelation function (ACF). The candidate ARIMA models are all possible combinations of p and q. The final ARIMA model is selected based on the Akaike Information Criterion (AIC).” were added to further explain the ARIMA model setup.

-The prediction of the loads should be compared with the observation to understand the model performance

Response: We think that this was done already and given in Figure 6 in the original submission.  Do we misunderstand the reviewer’s suggestion?

Other specific comments:

-L.33 2 space mg/L

Response: Thank you for pointing this out.  Correction is made.

-L.38 Add citation: Malago' et al. 2019 (Malagó, A.; Bouraoui, F.; Grizzetti, B.; De Roo,

Modelling nutrient fluxes into the Mediterranean Sea.J. Hydrol. Reg. Stud. 2019, 22, 100592)

Response: Thank you very much for recommending the article.  It is cited in this manuscript.

-L.64 Noncaptured? Uncaptured…but what do you mean? Wild?

Response: Thank you for your comments and suggestions.  “Wild” is used, instead.

-L.64-68 add references

Response: Thank you for your suggestions. References, “(Alexander et al, 2007; Robertson et. al., 2009)”, were are added

-L.80 Add more details about the main goals of the work and main methods applied….

Response: Thank you for your suggestions. We have added “.aims to reveal the patterns of annual variations, monthly variations of nutrient loads, and variations under extreme weather conditions” to the main goal.

-L.119 explain why you used the sum of No2 and No3. Add data entries and period of each variable in section 2.2. Here you should describe the observed time series not in the results (Figure 6 a and 7 a)…

Response: Only the sum of NO2 and NO3 load is given in the original data source available to us.  Therefore, we have to use the sum of NO2 and NO3 instead of separated loads.  We mentioned that the data covers a period of 19996 – 2016.  Figures 6a and 7a are associated with the ARIMA model development.  We prefer to leave these two figures where they are.

-Section 2.3. what do you mean with Net loads?  Rewrite this section adding

Method of calculation of loads Explain why you decide to calculate loads and not use directly concentrations Explain how have you calculated the trends as reported in line 170 I suggest to create a figure with the flow chart of work.

Response:

Net loads were the total of nitrites and nitrates.

-L.151-153 is method

Response: Thank you for the suggestion.  The sentence has been revised and moved to the Methods section.

-L.223-231 is method

Response: Thank you for the suggestion.  Some of the sentences are moved to the Methods section. However, we also keep a few other sentences in the place because they explain how the results are derived. We think it is better not to move these sentences to avoid abrupt change in the text flow.

-L.241 Some explanations of that?

Response: Thank you for your suggestion, explanation has been added.

-L.243-249 all the tests have to be described in the method!!

Response: We think it is better to keep this paragraph in the Discussion section because it is related to the data analysis and result discussion.

-L.271 and L. 331 all the equations should be explained in the method

Response: Thank you for your suggestion.  However, we prefer to keep the equations where they are so that the discussion can follow smoothly.

-the conclusions are very poor and not well supported by the results. I suggest to add a discussion section.

Response:

Thank you for your suggestion. The conclusion has been revised.

About Figures

-Fig.1 delete part a) and show the position of monitoring point.

Response:

Thank you for your suggestion. Figure 1a has been removed and the monitoring point was bordered by blue line.

-Fig.2 improve the figure and use consistent terminology OrthoP or PO4? Tons or t/yr…

Response:

Thank you for your suggestion. All the units and terminology have been changed.

Reviewer 3 Report

The results indicate 18 that (1) annually, the mean NO2-+NO3- and orthophosphate loads showed steadily increased during 1996- 1999, persistent level during 2000-2007, and increased again during 2008-2016;

This sentence is grammatically incorrect and needs to be re-written. I'm sure there are going to be more sentences like this, and suggest sending the article to an editor. The one I used charged $4 a page, but it is well worth it.

It would take more time than I have to correct all the errors that I expect and will just keep track of the number per page.

error

Line

33

34

I'm not going to be able to even count the errors. This needs editing by someone who knows English before it is sent out for review.  One time I reviewed an article written this poorly and that was enough.

Author Response

The results indicate that (1) annually, the mean NO2-+NO3- and orthophosphate loads showed steadily increased during 1996- 1999, persistent level during 2000-2007, and increased again during 2008-2016;

This sentence is grammatically incorrect and needs to be re-written. I'm sure there are going to be more sentences like this, and suggest sending the article to an editor. The one I used charged $4 a page, but it is well worth it.

Response:

Thank you for your suggestion, revision have been made.

It would take more time than I have to correct all the errors that I expect and will just keep track of the number per page.

error

Line

33

34

Response:

We made all appropriate corrections to the best of our ability. Thank you.

I'm not going to be able to even count the errors. This needs editing by someone who knows English before it is sent out for review.  One time I reviewed an article written this poorly and that was enough.

Response:

The document has been edited.

Author Response

This paper is limited in scope to water is covered. These aspects are slated to be studied at another time

Response:

Thank you for your suggestion

Review of manuscript [environments-622756] entitled Statistical Analysis of nutrient loads from Mississippi-Atchafalaya River Basin (MARB) to the Gulf of Mexico

By Okwan et al.

General comments

The authors present statistical analysis of nutrient loads on its treads, seasonality, etc for MARB. The manuscript is very well written and easily understandable. The topic itself and approach are of interest for journal publishing. Nevertheless, the scope of the manuscript needs to be expanded to enrich the scientific value of the paper (item 1-3 are ranked based on priority):

Particulate N/P (or TN, TP) should be included in the analysis. Particulate forms of N and P can actively contribute to hypoxia once reaching the bay and being decomposed by microbes. Moreover, this manuscript attempt to look at the effects of hurricanes on nutrient loads. Hurricanes enhance soil erosion and sediment transport, which means that the transport of particulate forms of N and P are more, or at least equally, impacted by hurricanes. The USGS dataset has TN and TP. I suggest the authors to make use of these data.

Response: Thank you for your suggestion.  However, the data used in this study are also from the U.S. Geological Survey report which only gives nutrient loads.  There is no particulate N and P concentration data given in the report.  We understand the reviewer’s point of view, but we were limited by the available data.  As the reviewer suggested, we will try to explore more in the USGS dataset in future.

The paper does not relate to the drivers for nutrient loading (sources) nor to hypoxic effect, but it concludes on both aspects (also in the abstract). I find it a heavy overselling. Meanwhile, I think including either of these would greatly increase of scientific value of the paper. Ideally, the authors should expand the scope of this study by including either the drivers (e.g., linking to the nutrient sources and agricultural activities) or the effect side (e.g., hypoxic extent). My suggestion is the latter as the data is readily available from the same USGS dataset.

Response: Thank you for your comments.  We revised the manuscript. As the purpose of this manuscript is not to study nutrient biogeochemical behavior and we are limited by the available data, we do not think we can do more to stretch too much.

The N:P ratio is an important driver for eutrophication, which is a contributing factor to determine the extent of hypoxia. The authors can add statistical analysis for the N:P ratio (trend and seasonality) to reflect on the impact of nutrient loading to hypoxia.

Response: Thank you for the suggestion.  The purpose of this study is to estimate and predict the nutrient loads under different conditions.  We did not attempt to study the nutrient biogeochemical cycle and organic matter diagenetic reaction.  Therefore, it may be better for us not to get into it in this manuscript.

Please rework on the conclusion section – it is too generic (see the relevant specific comments)

Response

Thank you for your suggestion. The conclusion has been edited.:

Comments on specific sections Abstract

Line 22: ANOVA: please add the full name

Response: Thank you for the suggestion.  It is done.

Line 23: the “normal/storm flow condition” here is misleading. Please be consistent with the terminologies in the main text (no hurricane, hurricane year)

Response:

Thank you for your suggestion. It has been edited.

Line 25-26: this sentence seems to suggest nutrient loads are higher in normal weather conditions than wet and dry conditions, but Line 22-24 highlights lower loads during normal flow conditions.

Response:

Thank you for your suggestion, the error has been corrected

Lines 20 – 22.  Analysis of variance (ANOVA) identified highly significant differences among seasonal loads; and (3) NO2-+NO3- and orthophosphate loads in normal flow condition were lower than those in storm flow condition.

Lines  23 – 24. Moreover, it is found that there is a significant (p < 0.01) increase in nutrient loads during the normal weather conditions So, you have to make it consistent.  I leave it to you.  (Huan)

Line 26-27: the influence of agricultural activities on nutrient loading is not analyzed at all in the paper. This should not be in the conclusion or abstract, unless the authors add the analysis (I advise the authors to do so).

Response: Thank you for the suggestion.  The words are deleted.

Section 1

Line 41-42: please rephrase. This statement is misleading. While this can be true for most of the developed world, for the developing world with poor sanitation and limited agriculture, this statement is not necessarily correct.

Please also check, e.g., Fink et al., 2018 (DOI:10.1002/2017GB005858), Seitzinger et al., 2010 (10.1029/2009GB003587)

Response:

Thank you for your comments.  We revised the sentences.  It now reads “According to a previous study (Du et al., 2018), agriculture has been considered as one of pollution sources of phosphates and nitrates.”

Line 42: what is “Industrial waste contamination”? if it is affected by municipal sewage (domestic not industrial contaminants), why is it industrial contamination? Please be careful with terminologies.

Response: Thank you for your comments  We have revised the sentences.  It now reads “In urban areas, domestic fertilize and detergent uses in the residential areas along the rivers or in the watersheds can also discharge nitrate, nitrite, ammonium, and phosphate directly into rivers orthrough municipal sewage run-offs, as a result of urbanization and population increase.”

Line 47-49: the literature to support the statement are outdated (15+ years old). Has the nitrate contamination changed in the study area in the recent decade?

Response:

Thank you for your observation, new references have been added

Line 66-67: the authors used present tense in the two statements. The SPARROW model was constructed in 1992. Are the statements still applicable for the situation of today? Also they contradict Line 75-77.

Response:

Yes. We provided references on the use of SPARROW model.

Line 72: yield -> yields

Response: Correction is made.

Line 79-81: the incentive/objective to conduct the study is vague. The paper does not relate to the drivers for nutrient loading (sources) nor to hypoxic effect. The authors need to be more specific on the objective of the study. Ideally, the authors should expand the scope of this study by including either the drivers (e.g., linking to the nutrient sources and agricultural activities) or the effect side (e.g., hypoxic extent). In either case, considering only NOx and orthophosphate is not sufficient.

Response: Thank you for your comments and suggestion.  This study was not designed to investigate the biogeochemical behavior of nutrients and the processes of hypoxia due to variations in nutrient input. This purpose of this study was to estimate the nutrient loads using available USGS data and investigate the impact of hurricanes.  We revise the text and made it clear now.

Section 2

Line 92 and 94: repeating texts. Revise.

Response: Thank you for your suggestion. Revised.

Line 119: why ammonium is not included if you have data for inorganic nitrogen (Line 116), which would include ammonium? Why particulate (organic) forms of N and P are not included in the analysis? They can actively contribute to hypoxia once reaching the bay and being decomposed by microbes.

Response: Thank you for your comments.  In our dataset, there is no information about dissolved oxygen concentration in the water body.  Therefore, we do not know the true condition in the water column.  The words “hypoxia” is misleading here.  The ammonium load is relatively low compared to (NO2+NO3) load, accounting for ~5%.  Because the purpose of this manuscript is to estimate the (NO2+NO3) and P load, we did not work on ammonium load data.  We do not have particulate N and P concentration data.

Line 116: where exactly are the sampling locations of the USGS data? At the outlet to the bay? Please specify and give more details on the dataset. The dataset seems to be (regression) modeling products and that should be mentioned as well.

Response:  We do not know the exact locations.  We obtained the data from the USGS report.  Thank you for your suggestion about the data processing.

Line 122-125: suggest moving the two sentences to section 2.1

Response:  Thank you for the suggestion.  The sentences have been moved to section 2.1.

Line 139-143: what is the purpose of the ARIMA forecasting model?

Response: We used the ARIMA forecasting model for the future prediction.

Section 3

Line 151-154: aren’t LOADEST and composite method used to estimate the USGS load dataset? If yes, shouldn’t this be in the data description section (2.2)?

Response: Thank you for your suggestion. A sentence in section 2.2 has been added on the use of composite data.

Line 159-169: 1. if the two methods gave different peak years, which method/data was used in the analysis afterwards? E.g., what is the data behind statements in Line 169-174? 2. Please add the plots for the time series with these two methods (can be as supplement materials) to support these statements.

Response: Thank you for your suggestion. The time series figures summarized in Figure 6 a – d.

Line 171-174: please add trend analysis to check whether the long-term (2-decade) trends are statistically significant.

Response: Thank you for your comment

Figure 3: Please specify whether these are annual, monthly or seasonal loads in the caption or y- axis (the numbers seem to suggest they are not annual loads); add legend for the boxplot – same for all other figures with boxplots.

Response: Thank you for your observation. The data presented in figure 3 was before and after hurricane seasons as shown in legend.

Line 194-196: p>0.05 is not significant. The statement contradicts Line 191-192. Is it an increase of mean OrthoP or NOx-? Figure3b shows a decrease of the mean OrthoP.

Response: Thank you for your observation. Corrected as P<0.05.

Figure 4: are these loads per season? Please specify. Same for Figure 5.

Response: The loads were based on seasons as shown in figures 4 and 5 (seasons before and after hurricane).

Line 216-217: what is the “timeline of development of hypoxia”? Although it’s obvious nutrient loading is a driver for hypoxia in the Bay, I am not convinced the seasonality of nutrient loading drives the seasonality of hypoxia. I’d rather argue that seasonal stratification (linked to wind speed/direct, temperature) and nutrient availability (nutrient loading + biochemical process in the bay area) have larger influence on the seasonality of hypoxia (e.g., Feng et al., 2019).

Response: Thank you for your suggestion. Hypoxia was not a part of this study. We made references of hypoxia in terms of the effect of N and P loads.

Line 221-222: I’d like to see the seasonality for hurricane occurrences and its association with the seasonality of nutrient loading.

Response: The data were summarized in figures 4 and 5 based on hurricane occurrences as listed in section 3.2.

Line 240-241: one explanation could be that hurricane carries particulates into the river systems, which lags more (lower velocity + instream mineralization) to deliver to the river mouth – another reason why particulate N/P should be included in the analysis.

Response: Thank you for your suggestion. This study does not include the variables you mentioned.

Line 347: the unit does not seem to be correct. Please double check.

Response: Thank you for your suggestion. The units were corrected as t/yr.

Line 351: 41.7±1,004.29 ton/year. Is the uncertainty/variation that high? Considering that the observed variation is rather small (44.3±10.5 ton/year), what is the point of the model if the uncertainty is that high?

Response: Thank you for your observation. It has been corrected.

Line 354: please reflect on the performance of the forecasting model. Can the model be used for future forecasting?

Response: Yes, we can use the model for future forecasting. A sentence is added during the revision, which reads “Therefore, this model has its significance to be applied in other regions with the similar settings.

Section 4 (Conclusions)

Overall, this section is too vague, generic, and in some cases not consistent with the results. The authors need to extract and draw the conclusions directly from the results.

Response: Thank you for your suggestion. The conclusion has been revised.

Line 358-359: this is not what Figure 4 shows for NOx.

Response: Thank you for your observation. It has been edited.

Line 359-361: if the authors want to conclude on the consistency in seasonality between hypoxia and nutrient loading, the discussion on the cause-effect relationship should be expanded and more in-depth.

Response: Thank you for your observation. It has been edited.

Line 361-363: this statement is too vague.

Response: Thank you for your observation. It has been edited.

Line 367: where are the impacts of agricultural activities mentioned in the result section? This statement is too far-reaching.

Response: Thank you for your observation. It has been edited.

Reviewer 5 Report

It is a good paper based on a rather simple but original work, which analyzed the observed nutrient data in the Mississippi river. The topic is important and deserves the attention of other scientists and of the readership in general. What makes it unique is the effort to identify differences between pre- and post-Katrina occurrence. I have the following comments which could increase the status of the paper:

1) Many references are included, some could certainly be skipped, while other very relevant papers related to the nutrient pollution of Mississippi do not appear. For example, the articles: https://doi.org/10.1016/j.jhydrol.2015.02.039 and  https://doi.org/10.1111/1752-1688.12594 could be cited in the paper. 

2) The English language needs some minor improvements. A careful reading by a native English speaker is suggested before resubmission. 

3) Be more accurate in Figures. Figure 1 seems old, while Figure 2 has no units on axes. Nutrient variables are not shown on Figure 4.

4) Some further discussion on the results is needed in this paper. Try to analyze them more analytically either in a separate section or within the results section.

Author Response

It is a good paper based on a rather simple but original work, which analyzed the observed nutrient data in the Mississippi river. The topic is important and deserves the attention of other scientists and of the readership in general. What makes it unique is the effort to identify differences between pre- and post-Katrina occurrence. I have the following comments which could increase the status of the paper:

Response: Thank you very much for your positive comments.

1) Many references are included, some could certainly be skipped, while other very relevant papers related to the nutrient pollution of Mississippi do not appear. For example, the articles: https://doi.org/10.1016/j.jhydrol.2015.02.039 and  https://doi.org/10.1111/1752-1688.12594 could be cited in the paper. 

Response:

Thank you for your suggestion. Added as suggested.

2) The English language needs some minor improvements. A careful reading by a native English speaker is suggested before resubmission. 

Response:

Thank you for your suggestions. The document has been edited.

3) Be more accurate in Figures. Figure 1 seems old, while Figure 2 has no units on axes. Nutrient variables are not shown on Figure 4.

Response:

Thank you for your observation. Units were added to axes in figure 4.

4) Some further discussion on the results is needed in this paper. Try to analyze them more analytically either in a separate section or within the results section.

Response:

Thank you for your suggestion, discussion has been included in the manuscript

Round 2

Reviewer 3 Report

The paper is still full of grammatical problems- which can all be addressed, I'm sure, but I can't get past them and don't want to spend my time correcting them. I suggested they hire an editor, and they didn't. Another suggestion is an online grammar checker, but if they have an english speaking co-author, that would be best if he read it and corrected it, which would show he made a contribution. Some journals require a major contribution to the paper by all authors, and I don't see how all of these authors could have contributed in a substantial way, especially, if they have not even read the paper.

Author Response

Thank you for your input. Per your advise we hired an editor to review and edit the manuscript.

Reviewer 5 Report

I think that after the revisions the paper has been improved and has the potential for publication. I have no other comments/suggestions apart from the recommendation to check the format in references. Be careful with the references added in the revised version. For example, ref 25 is a paper in the Journal of Hydrology, which however does not appear at the end of the reference. 

Author Response

Thank you for your observation. The references section has been revised.